# Modifying the Stability and Surface Characteristic of Anthocyanin Compounds Incorporated in the Nanoliposome by Chitosan Biopolymer

**DOI:** 10.3390/pharmaceutics14081622

**Published:** 2022-08-03

**Authors:** Mina Homayoonfal, Mohammad Mousavi, Hossein Kiani, Gholamreza Askari, Stephane Desobry, Elmira Arab-Tehrany

**Affiliations:** 1Bioprocessing and Biodetection Lab (BBL), Department of Food Science and Technology, University of Tehran, Karaj 999067, Iran; mina.homayoonfal@gmail.com (M.H.); mousavi@ut.ac.ir (M.M.); hokiani@ut.ac.ir (H.K.); iraskari@ut.ac.ir (G.A.); 2Laboratoire d’Ingénierie des Biomolécules (LIBio), Université de Lorraine, 2 Avenue de la Forêt de Haye, TSA 40602, CEDEX, 54518 Vandoeuvre-lès-Nancy, France

**Keywords:** chitosan, encapsulation, anthocyanin compounds, nanoliposome, encapsulation efficacy

## Abstract

In this study, a novel approach was investigated to improve the stability of anthocyanin compounds (AC) by encapsulating them in nanoliposomes resulting from rapeseed lecithin alongside chitosan coating. The results indicate that the particle size, electrophoretic mobility, encapsulation efficiency, and membrane fluidity of nanoliposomes containing anthocyanin compounds were 132.41 nm, −3.26 µm·cm/V·S, 42.57%, and 3.41, respectively, which changed into 188.95 nm, +4.80 µm·cm/V·S, 61.15%, and 2.39 after coating with chitosan, respectively. The results also suggest improved physical and chemical stability of nanoliposomes after coating with chitosan. TEM images demonstrate the produced particles were spherical and had a nanoscale, where the existence of a chitosan layer around the nanoparticles was visible. Shear rheological tests illustrate that the flow behavior of nanoliposomes was altered from Newtonian to shear thinning following chitosan incorporation. Further, chitosan diminished the surface area of the hysteresis loop (thixotropic behavior). The oscillatory rheological tests also show the presence of chitosan led to the improved mechanical stability of nanoliposomes. The results of the present study demonstrate that chitosan coating remarkably improved encapsulation efficiency, as well as the physical and mechanical stability of nanoliposomes. Thus, coating AC-nanoliposomes with chitosan is a promising approach for effective loading of AC and enhancing their stability to apply in the pharmaceutic and food industries.

## 1. Introduction

Barberry fruit is rich in phytochemicals with health-promoting effects, such as anticancer and anti-inflammatory properties [1]. Anthocyanin compounds (AC) in barberry, being responsible for the red color in its juice, are the dominant species of phenolic compounds present in this fruit, enjoying considerable antioxidant properties. Furthermore, they possess antimicrobial, vision-enhancing, and antihypertensive properties [2]. AC is the glycoside derivative of cation 2-phenyl benzopyran (flavylium), and like other natural flavonoids, they have the carbon skeleton of C_6_-C_3_-C_6_ [3]. AC are a group of polyphenolic flavonoids, the most abundant water-soluble pigments, extensively found in fruits, vegetables, and flowers.

Considering their known nontoxicity and biocompatibility with the human body, the use of AC as natural pigments and antioxidants has attracted a great deal of attention in recent years [4]. Nevertheless, since these compounds are unstable during food processes, distribution, storage, or in the gastrointestinal tract, their health-promoting potentials are limited. Further, temperature, pH, presence of oxygen, and enzymes can also affect the stability of anthocyanins [5]. Accordingly, adopting an effective strategy to improve the efficiency and stability of AC in their different process and environmental conditions is essential.

In recent years, encapsulation technology, which is an effective method for drug delivery in the pharmaceutical industry, has introduced itself as a useful means for improving and preserving the quality, stability, and bioavailability of bioactive compounds sensitive to external conditions [6]. Indeed, encapsulation is a process in which active agents are coated by carrier compounds, whereby particles or capsules at the micro or nanoscale are achieved. The encapsulation technique is widely used in food and pharmaceutical industries to coat bioactive compounds and create protective barriers against unfavorable environmental conditions [7]. In the final product of encapsulation, bioavailability, controlled release, and precise targeting of the bioactive compound improve. Recently, nanoencapsulation and development of delivery systems based on the nanoscale for bioactive compounds have grown interests because of advantages, including higher encapsulation efficiency, enhanced bioavailability, stability, and sustainable release profile [8].

Among various methods, the potential of liposomes as an encapsulation medium is proven in preserving, stabilizing, and delivering nutraceuticals [9,10,11,12]. Liposomes are colloidal vesicles, similar to biological membranes, and have a phospholipid structure. Liposomes are molecular associations as micro or nanospheres in whose structure one or several lipid bilayers are formed around an aqueous medium [13]. Due to special structural characteristics, both types of hydrophilic and hydrophobic molecules can be loaded in liposomes; hydrophilic compounds lie inside the internal water core, while hydrophobic compounds settle inside lipid bilayers [14]. Nanoliposomes protect sensitive bioactive compounds against different environmental conditions thanks to their amphiphilic nature, being biocompatible, biodegradable, non-toxic, and non-immunogenic. They can even enhance their bioavailability and absorption. Therefore, they have found extensive applications as ideal nanocarriers across different areas, including tissue engineering, and the pharmaceutical, food, and cosmetics industries [15].

Nevertheless, despite all of the mentioned advantages, nanoliposomes have the problem of a short half-life in the gastrointestinal tract [16]. Further, the incidence of processes such as degradation, aggregation, fusion, oxidation, and hydrolysis of phospholipids results in their physiochemical instability [17]. In recent years, modification of the surface structure of nanoliposomes by polymer coatings was regarded as a promising method to modify their efficiency and applicability. Considering the bio-polyelectrolytes, chitosan can be used as a polymer for coating the nanoliposomes and improving their efficiency [14].

Chitosan is a cation polysaccharide biopolymer composed of d-glucosamine units, obtained through deacetylation of chitin, the main component of the cellular wall of crustaceans, fungi, and insects. This hydrophilic compound is biodegradable and biocompatible, which is of interest as a mucoadhesive compound [18,19]. These characteristics cause better permeation of biomolecules through the barriers of the digestive system and the small intestine. Veritably, thanks to improving bio-adhesive characteristics and permeation, coating nanostructures based on lipids with chitosan results in the modified bioavailability of drugs through increasing their residence time at absorption sites [20]. Note that chitosan coating on nanoliposomes occurs through ionic interaction between amine groups with the positive charge in the chitosan and the diacetyl phosphate groups with the negative charge on the surface of liposomes [21].

So far, chitosan is used for coating nanoliposomes in various studies. Hassan et al. (2016) used chitosan for coating curcumin-containing nanoliposomes [22]. Gibis et al. (2016) employed this compound for coating nanoliposomes, including polyphenolic compounds [23]. Hao et al. used chitosan for coating nanoliposomes containing flavonoid quercetin [24]. Furthermore, various studies indicated that liposome systems could improve the physicochemical stability of AC. Accordingly, various methods, such as film hydration methods, combined with supercritical carbon dioxide, high-pressure, or ultrasonic homogenization techniques. Additionally, in most studies, synthetic or purified soybean phospholipids in combination with cholesterol are applied for AC-nanoliposome preparation [25,26]. However, there is not a reported investigation on the encapsulation of AC of barberry fruit in nanoliposomes coated with chitosan by hydration methods in combination with ultrasonic and high-pressure homogenization by employing a natural lecithin compound to improve the physicochemical stability and structural properties of AC nanoliposomes.

This study aims to encapsulate AC in nanoliposomes composed of rapeseed lecithin phospholipids as a natural source of lecithin and enhance the structural and physicochemical attributes of prepared nanoliposomes with chitosan coating. In this regard, the various mentioned characteristics of the produced samples were studied by size and surface charge of particles, TEM, FTIR, shear, and oscillatory flow behavior tests.

## 2. Materials and Methods

### 2.1. Materials

Fresh barberry fruits were prepared from Birjand, Iran. For the preservation of physicochemical properties, the fruits were dried at room temperature. Rapeseed lecithin was provided by an enzymatic extraction approach without applying any organic solvent. Chitosan (shrimps shell source, deacetylation degree up to 75%) was purchased from Sigma-Aldrich (Tokyo, Japan). Other chemicals include acetic acid, hexane, chloroform, ethanol, methanol, BF_3_/methanol, acetonitrile, 1-(4-Trimethylammonium-phenyl)-6-phenyl-1,3,5-hexatrien (TMA-DPH), sodium citrate, citric acid, potassium chloride, and sodium acetate, which were supplied from Sigma-Aldrich (Paris, France) and Fisher Scientific (Paris, France). All chemicals were of analytical grade.

### 2.2. Methods

#### 2.2.1. Fatty Acid Composition

To identify the fatty acid composition of phospholipids of rapeseed lecithin, first, the methyl ester of fatty acids was prepared. Then, separation of the different methyl esters was performed using a gas chromatography device equipped with a flame ionization detector (Perichrom, Saulx-lès-Chartreux, France). The temperature of the injector and detector was set at 250 °C [27]. The mixture of plant fatty acids was used to identify the fatty acids of rapeseed lecithin (PUFA from vegetable sources; Supelco, Sigma-Aldrich, Bellefonte, PA, USA). The results were presented as mean ± SD

#### 2.2.2. Lipid Class of Rapeseed Lecithin

The lipid class of rapeseed lecithin was determined by the Iatroscan device (Iatron Laboratories Inc., Tokyo, Japan). First, the samples were spotted on Chormarod (quartz wires with silica coating). Then, Chromarod was exposed to a mixture of solvents, including hexane, diethyl ether, and formic acid for 20 min, and then dried at 100 °C for 1 min. Eventually, the neutral lipids were detected by Iatroscan. In the next stage, the mixture of the polar solvent, including chloroform, methanol, and ammonia, was used for the identification of polar lipids. Then, internal software ChromStar was exerted for the interpretation of the results [28].

#### 2.2.3. Preparing the Barberry Anthocyanin Extract

Dry and cleaned fruits of barberry were used for preparing the AC extract. First, the fruits were mixed with ultrapure water at a ratio of 1:5, and then stirred using Ultra-Turrax homogenizer (T-25 Basic, IKA-Werk, Staufen, Germany) at 12,000 rpm for 5 min. In the next stage, the solid particles and impurities were removed using a vacuum filtration system and cellulose acetate with pore sizes of 10, 8, 3, 1, 0.45, and 0.2 µm, whereby a clear extract was obtained. Eventually, the extract was concentrated using a rotary vacuum evaporator up to the Brix 45°.

#### 2.2.4. Preparing Nanoliposomes Coated with Chitosan

To prepare nanoliposomes coated with chitosan, concentrated extracts of barberry and rapeseed lecithin were mixed with deionized water at ratios of 4.5 *w*/*v*% and 3 *w*/*v*%, respectively. The prepared mixture was stirred for 4 h under an inert atmosphere (nitrogen), and then chitosan (1 *w*/*v*%) alongside acetic acid (1 *w*/*v*%) was added to the prepared mixture. Again, the prepared mixture was stirred for 1 h under an inert atmosphere (nitrogen). It should be stated that the ratio of anthocyanin extract to rapeseed lecithin, as well as rapeseed lecithin to chitosan, were 1.5:1 and 3:1, respectively. In the next stage, to prepare a colloidal suspension of nanoliposomes, homogenization was performed in two stages. In the first stage, the samples were homogenized using an ultrasound homogenizer (probe with 13 mm diameter for the sample with 50 mL volume) at the frequency of 40 kHz and power of 40% for 4 min (1 s on and 1 s off). In the next stage, the samples were homogenized using a microfluidizer at the pressure of 1250 bar in eight consecutive cycles [22]. Finally, four species of nanoliposomes, including those devoid of chitosan coating and anthocyanin extract, LP-FR-UH, nanoliposomes containing anthocyanin extract without chitosan coating, LP-AC-UH, nanoliposomes devoid of anthocyanin extract with chitosan coating, LP-FR-CH, and nanoliposomes with anthocyanin extract and chitosan coating, LP-AC-CH, were produced and their characteristics were examined as follows.

#### 2.2.5. The Particle Size and Polydispersity Index of Nanoliposomes

The mean hydrodynamic diameter and polydispersity index (PDI) of nanoliposomes were examined through the Malvern Zetasizer Nano ZS device (Malvern Instruments, Malvern, UK) in dynamic light scattering (DLS) mode. First, the samples were mixed with ultrapure water at a ratio of 1:400. Next, they were placed inside vertically cylindrical covets with a diameter of 10 mm. Measurements were performed at the scattering angle of 173° and temperature of 25 °C. Further, the refractive index and extent of absorbance for the samples were considered as 1.471 and 0.01, respectively [29]. For all of the samples, the measurements were performed with three replications, as the results were reported as mean value ± SD.

#### 2.2.6. Electrophoretic Mobility

As with the size of particles, electrophoretic mobility of the nanoliposomes was measured by the Malvern Zetasizer Nano ZS device and electrophoretic light scattering (ELS) technique. First, the samples were mixed with ultrapure water at the ratio of 1:400 and then placed inside disposable capillary electrophoretic cells which contained copper electrodes by using a syringe [30]. The tests were performed at the temperature of 25 °C and the final result was reported as the mean value ± SD (four replications for each sample).

#### 2.2.7. Encapsulation Efficiency (EE)

To calculate the percentage of AC incorporated inside the nanoliposome structure, the samples were centrifuged for 1 h and 4 °C at 50,000 rpm, whereby two separate phases were obtained: the unencapsulated AC accumulated in the top phase, while the nanoliposomes sedimented in the bottom face. To remove the trace lecithin compounds, the top phase was mixed with chloroform at a 1:1 ratio and, after stirring, it was centrifuged for 10 min and 4 °C at 10,000 rpm. The concentration of AC in the top phase was calculated using the following Equation (1) [31,32]:(1)$$\mathrm{AC}\;\mathrm{concentration}\;\left(\mathrm{mg}/L\right)=\frac{\left[{\left(A_{510}-A_{700}\right)}_{\mathrm{pH}\;1}-{\left(A_{510}-A_{700}\right)}_{\mathrm{pH}\;4.5}\right]\times \mathrm{MW}\times \mathrm{DF}\times 1000}{\varepsilon \times L}$$

In the above equation, A_510_ and A_700_ are absorbances at the wavelengths of 510 nm and 700 nm, respectively, MW is the molecular weight of cyaniding 3-glycoside (449.2 g/mole), DF is the dilution factor, ε is the molar absorbance of cyaniding 3-glycoside (26,900), and L is the cell path length (1 cm). Absorbance was measured by a UV-Visible spectrophotometer (Varioskan^®^ Flash, Thermo Fisher Scientific Inc., Essone, France)

Furthermore, to calculate the total anthocyanin content, as mentioned earlier, the nanoliposomes were degraded by being mixed with chloroform, and their anthocyanin content was calculated [33]. Eventually, the encapsulation efficiency (EE) was measured using Equation (2).
(2)$$\mathrm{EE}\;\left(\%\right)=\frac{\mathrm{Total}\;\mathrm{AC}\;\mathrm{concentration}-\mathrm{unencapsulated}\;\mathrm{AC}\;\mathrm{concentration}}{\mathrm{Total}\;\mathrm{AC}\;\mathrm{concentration}}\times 100$$

#### 2.2.8. Membrane Fluidity

Membrane fluidity of nanoliposomes was measured using the method proposed by Hassan et al. [22] This method is based on determining the fluorescent level of the TMA-DPH compound. In this regard, the first TMA-DPH solution was prepared in ethanol and was added to samples such that eventually the concentration of 4 µM and 0.2 mg/mL^−1^ would be obtained for the probe and lipid, respectively. The resulting mixture, after 1 h of stirring in darkness and at the laboratory temperature, was poured inside black microplates (180 µL inside each well). Indeed, during this time, TMA-DPH and nanoliposomes react with each other, whereby fluorescence probes are distributed vertically and horizontally among the lipid layers. Determining the extent of fluorescence of the nanoliposome samples was performed using Tecan INFINITE 200R PRO (Austria) equipped with a fluorescent polarizer. Under constant stirring at 25 °C (environmental condition), the samples were excited and emitted at 360 nm and 430 nm, respectively. After data analysis by Magellan 7 software (Tecan Group Ltd., Mannedorf, Switzerland), the intensity of polarization of each sample (P) was calculated using the following Equation (3):(3)$$P=\frac{I_{∥}-{\mathrm{GL}}_{⊥}}{I_{∥}+2{\mathrm{GI}}_{⊥}}$$

In this relation, $I_{∥}$ and $I_{⊥}$ represent the extent of parallel and perpendicular fluorescent intensity to the excitation plane. G denotes the transmission factor, which is associated with the features of the device. The inverse of polarization, (1/P), was considered membrane fluidity.

#### 2.2.9. Transmission Electron Microscopy

The morphology and microstructure of nanoliposomes were studied via the negative staining method and by transmission electron microscopy (TEM). First, to decline the effect of concentration, the samples were diluted with distilled water about 30 times. Then, the diluted sample was mixed with ammonium molybdate solution 2% (as the negative staining agent) in a ratio of 1:1 and left exposed to room temperature. The resulting mixture was placed on copper grids coated with formvar-carbon for 5 min (mesh 200, diameter 300 mm). The additional samples were removed by filter paper and the grids were dried at room temperature. Eventually, the images of the stained samples were recorded by a Phillips CM20 transmission electron microscope with a voltage of 200 kV and using an Olympus TEM CCD camera [34].

#### 2.2.10. Fourier Transform Infrared (FTIR) Spectroscopy

FTIR spectra were achieved using a Tensor 27 mid-FTIR spectrometer (Bruker, Germany) provided with a DTGS (Deuterated Triglycine Sulfate) detector and a diamond ATR (Attenuated Total Reflectance) module. To determine the FTIR spectrum of the samples, the scanning rate was adjusted at a frequency of 20 kHz, in a such way that 128 scans were performed for each sample within the wavenumber range of 400–4000 cm^−1^ and resolution of 2 cm^−1^ at room temperature. Before each test, the reference spectrum (air) was recorded. Then, trace amounts of the lyophilized sample were placed on optical cell diamond crystal, and at least three separate tests were performed for each sample. Then, all of the obtained spectra were normalized by OPUS software (Bruker, Karlsruhe, Germany) [31].

#### 2.2.11. Rheological Characteristics

Rheological studies were performed by Kinexus pro rheometer (Malvern Instruments, Orsay, France). During the rheological tests, to minimize the changes in humidity, the measurement system and the sample were coated with a humidity chamber. Before each analysis, every sample was left in a rest state for 5 min, until reaching equilibrium conditions. The following tests were performed for each sample.

##### The Flow Behavior Tests

To measure the shear-steady viscosity and shear stress, cone and plane geometry (1°, 50 mm) was used. To determine the steady-state shear viscosity, the shear rate was increased within the range of 10^−3^–10^3^ S^−1^. Then, data of the shear stress against shear rate were fitted with the power-law model (Equation (4)).
(4)$$\sigma ={k\gamma }^{n}$$

In this equation, σ is the shear stress (Pa), k is the consistency index, and n shows the flow behavior index.

Furthermore, to estimate the plastic behavior of nanoliposome samples, the square root of shear stress was plotted against the square root of shear rate, whose data were fitted with a structure-based model, the Casson model (Equation (5)).
(5)$${\left(\sigma \right)}^{0.5}=k_{0C}+{\;k}_{c}\;{\left(\gamma \right)}^{0.5}$$

In Equation (5), Casson yield stress (σ_0C_) and Casson plastic viscosity (η_casson_) were calculated through σ_0C_ = (k_0C_)^2^ and η_casson_ = (k_c_)^2^, respectively.

Moreover, to evaluate the existence of dynamic hysteresis loss, the shear stress was first elevated from 10^−3^ Pa to 0.5 Pa and then decreased to the initial value within 30 min.

##### Oscillatory Rheology

To evaluate the mechanical properties of the nanoliposomes, first, the linear viscoelastic range of nanoliposomes was determined. In this regard, the shear strain range was increased from 0.1% to 1000% at a frequency of 0.5 Hz. Based on this test, the extent of strain lying within the linear range (10%) was considered for the frequency sweep tests (0.01–10 Hz). For this analysis, a cone-and-plane viscometer (diameter 20 mm with the inter-plane distance of 500 µm) was used. Then, variation of the storage (G′) and loss (G″) modules with the elevation of frequency was measured at constant strain. The oscillatory rheological tests were measured at 25 °C.

## 3. Results and Discussions

### 3.1. Composition of Fatty Acids

The results of the analysis of fatty acids of rapeseed lecithin indicate that monounsaturated fatty acids were the prominent types where polyunsaturated and saturated fatty acids claimed 35.09 and 6.14% of the total content of fatty acids, respectively. Meanwhile, the most abundant fatty acids of rapeseed lecithin belonged to oleic acid, linoleic acid, and palmitic acid, respectively.

### 3.2. Lipid Class

The evaluation of the lipid class of rapeseed lecithin using the thin layer chromatography method showed that the content of neutral and polar lipids of rapeseed lecithin was 73.96% and 26.31%, respectively. Further, phosphatidylcholine, phosphatidylethanolamine, and phosphatidylinositol were the constituent phospholipids of rapeseed lecithin.

### 3.3. The Particle Size and PDI

The results of particle size of the nanoliposomes prepared using the combined techniques of sonication and microfluidization are presented in Table 1.

As can be observed, the mean particle size of LP-FR-UH and LP-AC-UH nanoliposomes was 121.60 nm and 132.41 nm, respectively. On the other hand, after coating with chitosan, these values increased to 261.5 nm and 188.94 nm for LP-FR-CH and LP-AC-CH samples, respectively. The elevation of particle size is due to the interactions between lipid and chitosan: indeed, chitosan, a copolymer consisting of N-acetyl-d-glucosamine and d-glucosamine, because of the presence of amine groups, has a positive charge in acidic environments. Owning to this special chemical characteristic, chitosan can interact with negatively charged compounds or those rich in electrons [35,36]. Therefore, it can be concluded that the electrostatic interaction between the groups of the head of phospholipids and special functional groups of chitosan, NH_3_^+^, as well as the hydrophobic interaction between the hydrocarbon chain of lipids and chitosan, result in the development of chitosan layer around nanoliposomes, and its entrance to the bilayer part of nanoliposomes eventually leads to the increment of the particle size of coated nanoliposomes [37]. Although the mean particle size of LP-AC-CH nanoliposomes was larger than that of LP-AC-UH samples, the notable point was the diminished size of the particles of these samples in comparison to LP-FR-CH ones. So far, chitosan has been used in different studies for nanoliposome coating [22,24,38]. All studies suggested an increased diameter of polymer-coated nanoliposome particles, along with the addition of bioactive or pharmaceutical compounds, which is incongruent with the results of the present study. This is probably due to the interaction between chitosan, anthocyanin, and phospholipids. Indeed, due to the existence of hydroxyl groups in the structure of anthocyanin compounds, phosphate groups within the structure of nanoliposomes, and hydroxyl as well as amine groups in the structure of chitosan, the probability of the formation of hydrogen and electrostatic interactions between these compounds increases [22,25,31]. The extent and intensity of these attraction forces are such that lead to compression of the nanoliposome structure or reduction in particle size in comparison to the unloaded samples. During the conversion of layered structures to vesicles, these compounds settle among the created molecular pores. Possibly, the concurrent addition of chitosan and anthocyanin extract to the system results in increased rigidity of the wall of vesicles, and thus diminished coalescence and particle size of the system. Furthermore, the existence of the mentioned compounds results in increased density of the arrangement of phospholipid molecules, thereby reducing the particle size [39].

The results obtained from the distribution index of nanoliposomes are also shown in Table 1. The results suggest that all of the studied systems had a PDI less than 0.258 with a monomodal peak. For the colloidal dispersion systems, a PDI less than 0.3 was ideal, suggesting the smallness of the width of the particle size distribution diagram [28]. Accordingly, the systems enjoy acceptable stability. The results also show that the samples containing AC and chitosan had a PDI less than the samples lacking AC, and the particle size diagram width is smaller in them. This might be due to the complex interaction between phospholipids, chitosan, and AC, as mentioned earlier. Indeed, the outcome of attraction interactions in a three-component system results in increased compression in the structure of nanoparticles, improved homogeneity and uniformity, and eventually reduction in PDI value.

### 3.4. Electrophoretic Mobility

To identify the extent of electrostatic repulsion between nanoparticles and resulting system stability, the electrophoretic mobility of the samples was evaluated with its results presented in Table 1.

Electrophoretic mobility is an index that is evaluated to investigate the physical characteristics of colloidal drug delivery systems. This characteristic represents the surface charge of particles and is one of the important parameters which affect the behavior of nanoliposomes. The magnitude of electrophoretic mobility is a reason for the stability of colloidal systems. Indeed, when the magnitude of this index grows, repulsion between particles increases; thus, the tendency of particles to aggregate with each other diminishes, causing greater stability of colloidal dispersions [28,40]. As shown in Table 1, the absolute value of the electrophoretic mobility of all samples was larger than 3. This suggests that, due to high electrostatic repulsion among the nanoparticles, the samples enjoy acceptable kinetic stability. The samples without chitosan had negative electrophoretic mobility due to the presence of anionic groups of phosphate in the structure of phospholipids, which altered into positive after the addition of chitosan to the colloidal system. These results were also reported in studies by other researchers who used chitosan as a secondary coating in the encapsulation process [14,41]. The final value of electrophoretic mobility for the LP-FR-CH and LP-CH-AC samples was +4.51 µm·cm/V·S and +4.80 µm·cm/V·S, respectively. The nanoliposomes containing chitosan form in response to different bonds, including van der Waals, hydrogen, and electrostatic, among which electrostatic interactions between positively charged polyelectrolyte and negatively charged surface groups of nanoliposomes attract them towards each other, causing the formation of a thin layer in the surface [15,22]. Changes in the electrophoretic mobility of samples from −3.26 µm·cm/V·S to +4.80 µm·cm/V·S were due to the absorption of cationic groups of chitosan polymer across the nanoliposome surface, suggesting that the nanoparticles have been well coated by chitosan.

### 3.5. Encapsulation Efficiency

The encapsulation efficiency (EE) was measured for the AC entrapped in nanoliposomes with and without chitosan coating. The results indicated that the application of chitosan resulted in improved EE, elevating it from 42.57% to 61.15% (Table 1).

Similar results have also been revealed in various studies. Hassan et al. (2016) as well as Gibis et al. (2014), found that the application of chitosan for coating nanoliposomes led to enhanced EE [22,38]. Generally, EE depends on the physicochemical characteristics of the constituent materials of the wall and core of nanocapsules. Nanoliposomes have a water core and a very thin hydrophobic layer [22]. During the encapsulation process, chitosan molecules settle on the surface or insight the bilayer part of phospholipids (filling in hollow spaces), causing diminished leakage of anthocyanin compounds from nanoliposomes, causing elevated encapsulation efficiency to 30.38%.

### 3.6. Membrane Fluidity

The results of the effect of AC and chitosan on the membrane fluidity of nanoliposomes are shown in Table 2.

As demonstrated, LP-AC-UH samples had less membrane fluidity compared to LP-FR-UH. Further, LP-FR-CH samples had less membrane fluidity compared to LP-FR-UH. Eventually, nanoliposomes LP-AC-CH had less membrane fluidity compared to LP-FR-CH. Basically, the number of bilayers constituting the membrane and its fluidity are the features significantly affecting the release of bioactive compounds from nanoliposomes. The most important influential factor on membrane fluidity is the liposome constituent components and their interactions with other compounds present in the system [42]. However, the curvature in the structure of unsaturated fatty acids causes their interactions with other lipids to be less strong, thereby enhancing membrane fluidity [43]. Therefore, the arrangement of external compounds in the phospholipid bilayer leads to limitations in the random motions of the head groups of lipids and eventually diminished membrane fluidity. Admittedly, chitosan polymer forms a new layer around the liposome, then interacts with the phospholipid bilayer membrane, causing the strength of the lipid bilayer, diminished freedom of motion of fatty acid chain and phosphate groups, and, eventually, membrane fluidity.

### 3.7. Transmission Electron Microscopy (TEM)

The appearance and morphology of the nanoliposomes with and without chitosan were evaluated by the TEM technique and the results are shown in Figure 1.

As illustrated, the size of both groups of nanoliposomes lay within the nanoscale. Also, TEM images suggest that the nanoliposomes prepared through the ultrasound and microfluidization combined method are in the form of spherical vesicles. Moreover, the morphology of chitosan-coated nanoliposomes indicates the existence of a layer around the vesicle, while no damage is observed in the membrane wall after coating [22,31,40].

### 3.8. Stability of Nanoliposomes

The results of investigating the stability of nanoliposomes indicated that all samples, at the studied temperatures, enjoyed suitable stability over time, and there was no significant difference between the studied parameters compared to the first day. Only in the LP-AC-UH sample was a significant decline of EE observed on the 30th day and 37 °C, where the EE decreased to 37.84%. However, the EE of LP-AC-CH samples at the end of storage time and both 4 and 37 °C did not show any significant difference compared to the initial value. Fundamentally, the release of bioactive compounds from nanoliposomes is largely dependent on lipid membrane fluidity [44]. Incorporation of AC along with the nanoliposome lead to an organized arrangement of phospholipid membranes, an increase in their packing, and finally a diminished membrane fluidity (Table 2). Indeed, chitosan chains settle in the external and internal region of the nanoliposome membrane, thereby organizing the membrane bilayer structure and preventing the release of encapsulated compounds [24,45].

### 3.9. Rheological Characteristics

#### 3.9.1. Flow Properties

In steady-state rheological tests, the flow behavior of nanoliposome samples with different formulations was evaluated. Also, the obtained data were fitted to power law and Casson models (Figure 2), and the parameters related to these models are presented in Table 3. As shown (Figure 2a), the shear stress had virtually a linear dependence on the shear rate for LP-FR-UH and LP-AC-UH samples.

However, the addition of chitosan to the colloidal systems led to changes in the flow behavior. As presented in Table 3, based on the k value, the samples with chitosan coating had higher apparent viscosity compared to samples without it. Furthermore, concerning n values, the samples without and with chitosan revealed Newtonian and shear thinning (pseudoplastic) behavior, respectively.

Different studies have reported that colloidal dispersions generally show rheological behavior within the Newtonian to pseudoplastic behaviours [46,47,48]. According to Einstein’s equation, the viscosity of dispersions is directly proportional to the viscosity of the continuous phase, where the rheology of colloidal systems is more affected by the characteristics of this phase. In this study, water was the dominant component of the dispersion of nanoliposomes. Further, based on the value of the consistency index of LP-FR-UH and LP-AC-UH samples, it can be inferred that the viscosity of these samples is similar to water (0.001 Pa·S). In these samples, the constancy of viscosity with the elevation of shear rate suggested that the system contains small particles which have not been coagulated, implying system stability [46,47,49].

The presence of chitosan on the one hand leads to enhanced viscosity of the system (k value: 0.027) by affecting the properties of the continuous phase. On the other hand, the interactions of the crystalline network of the lipid phase and the biopolymer can also affect the rheology of the colloidal dispersion. The attraction forces between lecithin and chitosan encourage the formation of the chitosan layer and enhance the viscosity of the system [50,51].

The emergance of the shear thinning behavior is because of coalescence or the existence of thickening compounds in the system. In the latter case, reduction in viscosity with the elevation of shear rate is associated with degradation of intermolecular interactions of the polymer and colloidal system components that eventually result in the increased mobility of chitosan chains and pseudoplastic behavior. On the other hand, applying shear forces leads to the release of the continuous phase (water) entrapped in chitosan chains, thereby thinning the dispersion and reducing its viscosity [52,53].

As shown in Table 2, for all of the four colloidal systems under study, stress-strain experimental data were properly fitted with the Casson model (R^2^ > 0.9846), representing the plastic behavior of the samples. Therefore, yield stress above zero suggests resistance of the samples to the external force before being flowed [54]. In this series of tests, yield stress and plastic viscosity increased after the coating of nanoliposomes with chitosan. The magnitude of yield stress is associated with the strength of interactions among particles in the three-dimensional microstructure network of nanoliposomes. Therefore, as confirmed by the results, the presence of chitosan in the system leads to better stability of vesicles in the rest state through interacting with other components (anthocyanin compounds and lipid structures).

Furthermore, the time dependency of the flow behavior of nanoliposome samples has been shown in Figure 3a. As illustrated, all of the samples manifested thixotropic behavior. Nevertheless, the area of the hysteresis loop in the samples with chitosan decreased significantly, suggesting the high stability of these samples. Indeed, the nanoliposomes coated with chitosan had less thixotropic characteristics and thus greater ability in repairing their structure after resolving the shear forces. The hysteresis loop area resulting from the rheopectic behavior (without the presence of chitosan) is possibly due to the dissociation of vesicles in response to shear. Veritably, the chitosan layer developed around nanoliposomes leads to the stability of vessels, preventing their deformity in response to shear forces [55,56].

Figure 3b demonstrates the viscosity of samples at very low shear rate values in a such way that all samples lacked Newtonian behavior. It is also observed that the zero shear viscosity of the nanoliposomes increased from 7.27 Pa·S to 27.01 Pa·S after coating with chitosan. This suggests that at values less than the critical shear rate or yield stress, there is infinite resistance against the flow. Indeed, when the shear rate increases enough to overcome the Brownian motion, the system particles are positioned along the field, where they will have less resistance against the flow and naturally less viscosity [57].

#### 3.9.2. Oscillatory Shear Characteristics

The results of the frequency sweep tests (Figure 4) reveal that across all of the LP-FR, LP-AC, LP-FR-CH, and LP-CH-AC samples, with the elevation of the frequency, the storage module, and viscose module grow. The storage module (G′) is a criterion for the extent of stored and recovered energy in each cycle of deformity (force exertion), which reflects the pseudo-solid component of viscoelastic behavior. On the other hand, the viscose module (G″) indicates the extent of energy lost in each cycle and describes that pseudo-liquid component. In this study, all samples at low frequencies had G″ values higher than G′, and showed pseudo-liquid behavior. In this estate, the energy required for deformity of the materials has been dissipated as a viscosity [54]. As illustrated in Figure 4, in all samples, gradually with the elevation of frequency, G′ and G″ curves cross each other, whereby the tendency to pseudo-solid behavior increases. In the study, this was observed for the samples without chitosan and those with chitosan at the frequencies of around 0.4 Hz and 0.03 Hz, respectively. These observations represent the greater ability of chitosan-containing samples in the formation of gel [53]. As shown, the viscoelastic behavior of lipid vesicles was not affected by the encapsulated anthocyanin extract, while the effect of chitosan on these characteristics is evident. The presence of chitosan in the system has resulted in the elevation of G′ and G″ and eventually improved the mechanical properties of nanoliposomes [54]. Various factors, including the nanoparticle size, mechanical properties of the chitosan layer, and their interactions, can cause improved stability of liposomes with a secondary coating. As mentioned earlier, chitosan acts as a protective coating and decreases phospholipid layer deformity. Typically, the mechanical stability of liposome dispersions is affected by different interactions of lipid vesicles, including van der Waals, electrostatic repulsion, and long-range entropy repulsion. When the lipid bilayer systems have chitosan coating, the above-mentioned interactions are intensified contributing to mechanical stability [58].

### 3.10. FTIR Analysis

FTIR analysis was performed as a non-destructive technique for identifying and analyzing the functional groups of nanoliposomes of rapeseed lecithin, chitosan, and anthocyanin extract, and their interactions within the wavenumber ranges of 440–4000 cm^−1^. The results of these tests are shown separately in Figure 5 and Figure 6.

In the FTIR spectrum of chitosan, as seen in Figure 5, the peak observed within the range 3000–3700 cm^−1^ is related to stretching bonds of hydroxyl as well as symmetric and asymmetric stretching bonds of N-H in the amine group. The vibrations of carbonyl bonds (C=O) of amide group CONHR (secondary amides) were recorded at 1631 cm^−1^ as well as the protonated amine groups, NH_3_^+^, at the wavenumber of 1543 cm^−1^. On the other hand, the absorption bands belonging to 852–1203 cm^1^ are attributed to the saccharide structure. Further, the peaks at 1022 cm^−1^ and 1066 cm^−1^ are related to C-O stretching vibrations [59,60].

Figure 5 demonstrates the FTIR spectrum of the main bands constituting phospholipid vesicles. The wideband present within the wavenumber range of 3045–3722 cm^−1^ represents the presence of OH. The peak emerging at 1743 cm^−1^ shows the stretching vibrations of carbonyl ester groups of phospholipids (-C=O). The absorption bonds belonging to stretching vibrations of double carbon–carbon bonds of alkenes, C=C were detected at 1651 cm^−1^. The symmetric and asymmetric vibrations of PO_2_^−^ were revealed at 1224 cm^−1^ and 1081 cm^−1^. Further, the bands present at 1064 cm^−1^ and 1163 cm^−1^ are assigned to symmetric and asymmetric stretching bonds of CO-O-C. Eventually, the bands at 975 cm^−1^ represent the asymmetric stretching vibrations of N(CH_3_)_3_ [61,62].

In the FTIR spectrum of anthocyanin extract (Figure 5), the bands at 600–1045 cm^−1^ indicate the presence of an aromatic ring, while 1028 cm^−1^ is related to the transformation of the C-H aromatic ring. The 1051 cm^−1^ is due to the stretching vibrations of C-O-C esters, while the peak at 1259 cm^−1^ is associated with the stretching of the pyran ring, the typical structure in flavonoid compounds. The band at 1400 cm^−1^ is related to the transformation of C-O, 1589 cm^−1^, and 1701 cm^−1^ allocate to C=C and C=O groups of the aromatic ring, respectively. The wideband at 2985–3710 cm^−1^ is related to the O-H group of phenol and sugar vibrations [59,63,64].

After coating nanoliposomes with chitosan, the interaction between polysaccharides and lipids was examined by FTIR. The FTIR spectrum of phospholipid–chitosan nanoparticles is presented in Figure 6. The results indicate that the formation of the complex between the liposome and chitosan resulted in changes in the range and intensity of absorption bands of the OH groups. This is due to the presence of more hydroxyl groups and the probability of the development of new intermolecular hydrogen bonds [65]. Phosphate groups are among the functional groups present in the structure of nanoliposomes being able to form bonds with amino polysaccharide (chitosan) through electrostatic interactions. The results also suggest that the absorption bands of phosphate groups of liposomes shifted to a higher frequency (from 1224 cm^−1^ to 1232 cm^−1^), suggesting that the interaction with polymer ligands resulted in the dehydration of phosphate groups. Indeed, the characteristic bands of chitosan at 1543 cm^−1^ related to scissoring vibrations of the NH_3_^+^ protonated amine group, shifted to a higher frequency at 1554 cm^−1^. These changes suggest that most hydrogen bonds of the phosphate group were broken due to electrostatic interaction with the chitosan amine group [22]. The carbonyl group is another site that has the potential to attach to chitosan. This group has a partially negative charge on the oxygen atom. The interaction between liposome and chitosan results in considerable changes in the region related to absorption of the carbonyl group and transfer of absorption bands to higher frequencies (from 1732 to 1755 cm^−1^). This absorption band position represents a considerable decline in hydration groups. These results suggest the degradation of several hydrogen bonds due to the involvement of carbonyl groups with the cationic group of the polymer, similar to the phosphate group of nanoliposomes. Thus, these observations suggest that both carbonyl and phosphate of liposome act as sites for establishing bonds with chitosan. The results also indicate that the absorption bands belonging to the carbonyl amide group, wave number 1631 cm^−1^, do not exist in the LP-FR-CH sample. These observations are possible since this functional group merges with carbonyl groups of phospholipids, leading to the elimination of the peak at 1631 cm^−1^ [22,41,62].

The FTIR spectrum related to LP-AC-UH samples is also shown in Figure 6. As can be seen, various peaks are observed in the absorption spectra related to AC and liposomes. Some of the peaks related to liposome and anthocyanin extract remain unchanged, while some undergo some variations. The position of the carbonyl group of lipids after attachment of anthocyanin compounds changed and shifted from 1743 cm^−1^ to 1730 cm^−1^. However, the phosphate group of lipids changed from 1224 cm^−1^ to 1240 cm^−1^. In addition, the appearance of the region related to O-H groups is also altered. The restriction of bands of O-H is associated with the development of hydrogen bonds. The OH peak of AC shifted as much as 17 cm^−1^ towards lower wavenumbers in the presence of liposomes. This might be due to the breakdown of hydrogen bonds and the formation of hydrogen bonds between OH groups of anthocyanins (phenolic and sugar) and functional groups of phospholipids (phosphate and carboxyl) [59]. Popova et al. (2016) concluded that amphiphilic phenolic compounds prefer to lie at the interface of water–lipid of lipid bilayers and close to P=O and C=O groups, such that their longitudinal axis would be arranged in parallel to the lipid bilayer plane. In this state, the OH groups of sugar ring establish with P=O gropes, and phenolic OH groups create hydrogen bonds with C=O groups. Since the phenolic OH group enjoys less flexibility in orientation, the aglycon structure thus plays a more important role in establishing hydrogen bonds with C=O groups as compared to P=O [59]. The peaks between 1498 cm^−1^ and 1662 cm^−1^ are related to the phenolic ring present in the structure of anthocyanins. The peak at 1701 cm^−1^ associated with C=O vibrations is influenced by the presence of lipids and transfers to lower frequencies. This suggests that the carbon group present in the phenolic C ring of anthocyanins established a hydrogen bond with lipid molecules. As can be observed, at 1375 cm^−1^, a new peak is created with a low absorption intensity in the structure of nanoliposomes containing anthocyanin extract. This absorption band may belong to -N=O stretching vibrations. Indeed, N(CH_3_)_3_ group present in the lecithin structure is highly electrophilic and contains an empty orbital. This group forms a weak Dative bond with the free electron pair of the OH group in the structure of anthocyanins. Then, due to the resonance effect of π electrons of the benzene ring, it temporarily changes into a weak -N=O bond, creating a new peak at 1375 cm^−1^ [66].

There is another probability for the emergence of a 1375 cm^−1^ peak. The absorption bands 1259 cm^−1^ in anthocyanins (C-O aromatic ring) and 1228 cm^−1^ (PO_2_^−^ asymmetric stretching) in lipids in the final compound have changed into 1240 cm^−1^ and 1375 cm^−1^ [67]. Other probabilities can be considered for this event. Indeed, the ether C-O bond has a positive charge and empty orbital and tends to create a bond with negatively charged compounds. On the other hand, the band 1228 cm^−1^ belongs to phosphates and has a negative charge. Possibly, the interaction between the two mentioned functional groups results in their integration. Nevertheless, the reaction of the ether group of anthocyanin Pyran ring is not exclusive to the phosphate group, and it can create a reaction with other groups with a negative charge. This also applies to the phosphate group [59]. Overall, the reactions that may occur for both functional groups with result in shifts in their peaks and overlap of the created peaks.

It is also observed that the symmetric stretching peaks belonging to CO-OC functional groups and PO_2_^−^ in the IR spectrum of nanoliposomes do not exist in LP-AC-SH. Further, the peak belonging to the asymmetric stretching vibrations of CO-OC has shifted to 1180 cm^−1^. This can be attributed to the fact that both groups have a potential for creating hydrogen bonds with the OH groups inside the anthocyanin extract structure, where the intensity of development of interactions has resulted in displacement or disappearance of peaks [22].

Figure 6 indicates the FTIR spectrum related to the interaction between chitosan and anthocyanin. As can be observed, the 1631 cm^−1^ peak associated with the carbonyl band of the CONHR amide group and the peak belonging to the carbonyl group of the AC aromatic ring at 1703 cm^−1^ do not exist in the final sample. These results suggest the development of a hydrogen bond between chitosan and anthocyanin [22,31].

On the other hand, the peak related to scissoring vibrations of NH^+^_3_ in chitosan and the peak associated with C=C vibrations of the aromatic ring do not exist in the final sample. However, within the range of 11,134–1803 cm^−1^, peaks with 1404 and1562 cm^−1^ centrality have been formed. Indeed, the main reason for these changes is the participation of the mentioned groups in hydrogen interactions, whose result is possibly a shift of these peaks or their integration with each other.

## 4. Conclusions

In this study, for the first time, a novel method of encapsulating AC using rapeseed lecithin and chitosan was investigated. It is indicated that AC was successfully encapsulated inside nanoliposomes coated with chitosan. Further, in the present study, the physicochemical and structural characteristics of nanoliposomes, and nanoliposomes with chitosan coating before and after encapsulation of anthocyanin compounds were conducted. Elevation of particle size and surface charge of liposomes suggested changes in the surface characteristics of nanoliposomes after coating with chitosan through the interaction between polymer and liposomes. Also, TEM images confirmed the existence of the chitosan layer around nanoliposomes. Diminished membrane fluidity in the presence of anthocyanin compounds and chitosan suggested decreased mobility of fatty acids chain, causing reduced leakage of target compounds to the outside of nanoliposomes. Rheological studies and flow behavior of nanoliposomes showed that the presence of chitosan leads to altered behavior from Newtonian to pseudo-plastic, diminished thixotropic behavior, and enhanced stability of colloidal dispersions of nanoliposomes. Also, oscillatory rheological behavior suggested greater mechanical stability of nanoliposomes samples with chitosan coating. The results of the FTIR test revealed that the interactions between nanoliposomes and chitosan included electrostatic (between cationic ammonium groups of chitosan with anionic phosphate groups of nanoliposomes), hydrogen, and hydrophobic. The same interactions occur between AC and liposomes, with the only difference being that the electrostatic interaction occurs between the oxygen of ether ring of anthocyanin with a positive charge and phosphate group with the negative charge of nanoliposomes. However, the interaction between anthocyanin and chitosan is only of hydrophobic and hydrogen types. In conclusion, it can be stated that coating AC-nanoliposomes with chitosan enhances their structural characteristics and the physicochemical stability of ACs, which can be a promising strategy for utilizing such compounds in health-promoting products. In future studies, the aim will be to evaluate the release behavior of anthocyanin compounds from chitosan-coated or uncoated AC-liposomes in the simulated gastrointestinal tract.

## Figures and Tables

**Figure 1 pharmaceutics-14-01622-f001:**
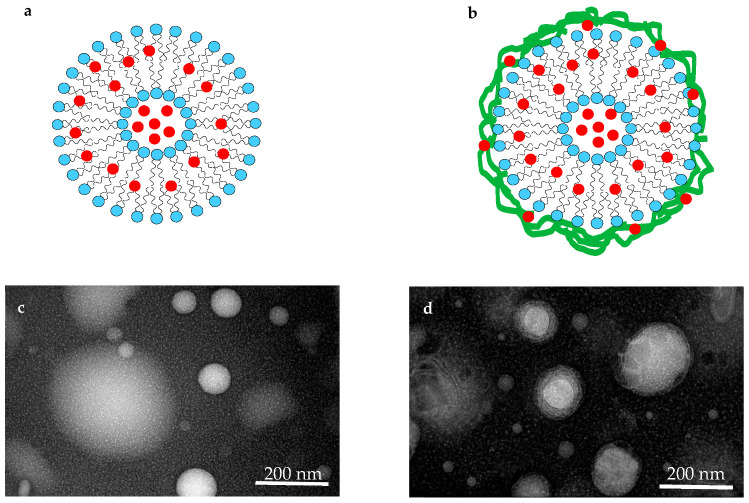
Schematic illustration of nanoliposomes: (**a**) uncoated nanoliposomes; (**b**) coated nanoliposomes; TEM images of nanoliposomes; (**c**) uncoated nanoliposomes; (**d**) chitosan-coated nanoliposomes. 
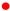
: Anthocyanin compounds, 
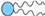
: Nanoliposome, 
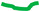
: Chitosan.

**Figure 2 pharmaceutics-14-01622-f002:**
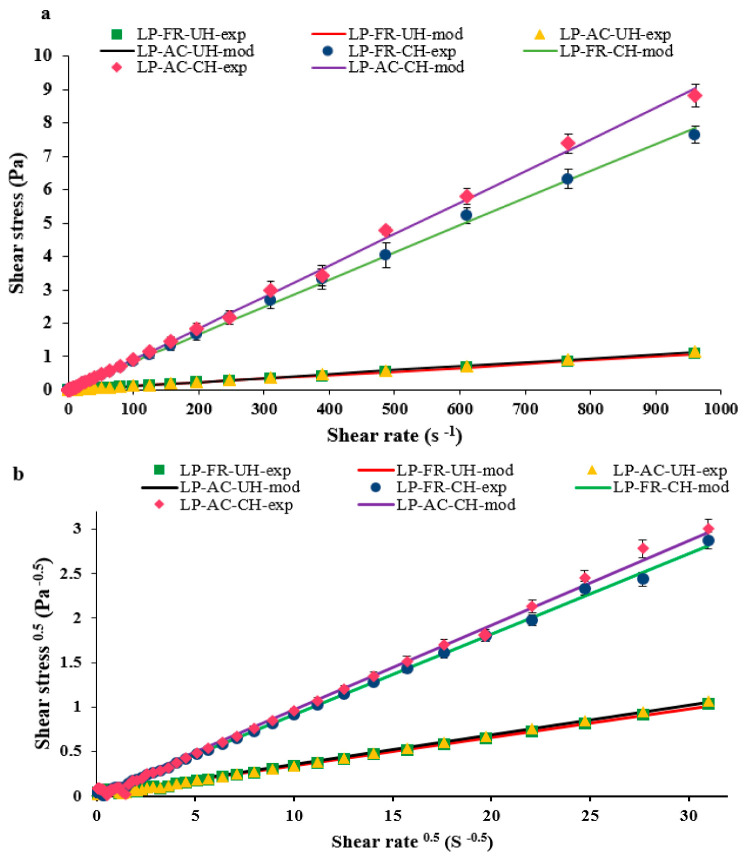
Flow behavior of nanoliposomes with various formulations: (**a**) Flow curves of shear stress and fitted data with the power-law model; (**b**) experimental data fitted with Casson model (square root of shear stress against the square root of shear rate).

**Figure 3 pharmaceutics-14-01622-f003:**
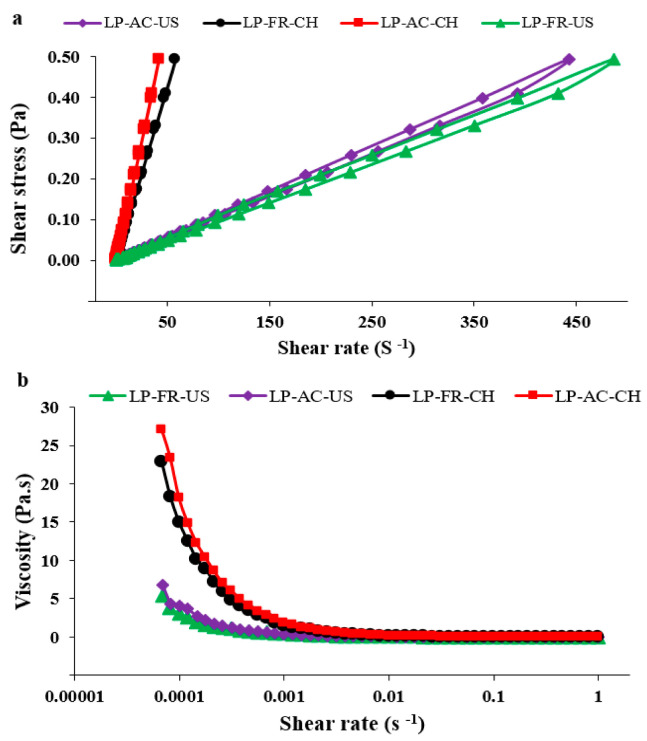
(**a**) Time dependency of the flow behavior of nanoliposomes with various formulation; (**b**) Shear viscosity at low shear rate.

**Figure 4 pharmaceutics-14-01622-f004:**
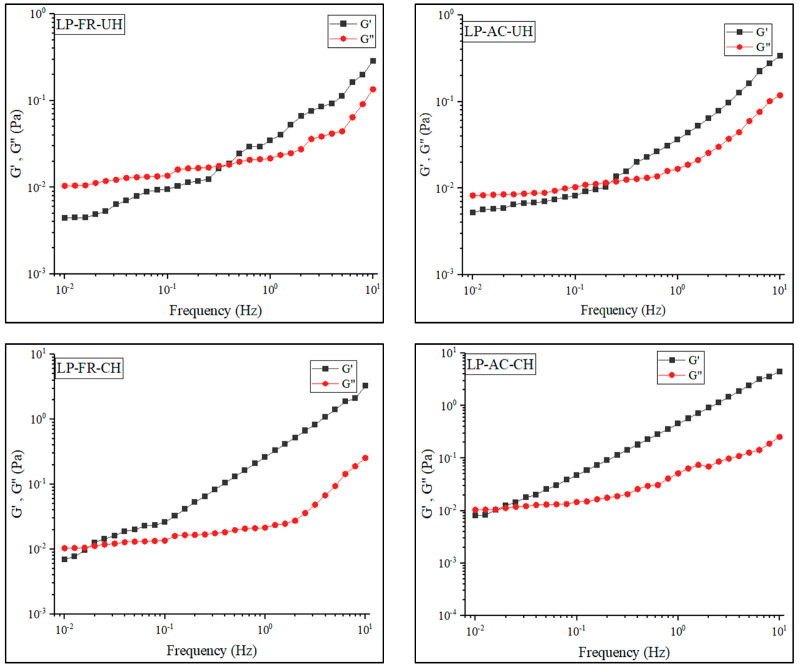
Frequency sweep of different nanoliposome samples at strain 10%.

**Figure 5 pharmaceutics-14-01622-f005:**
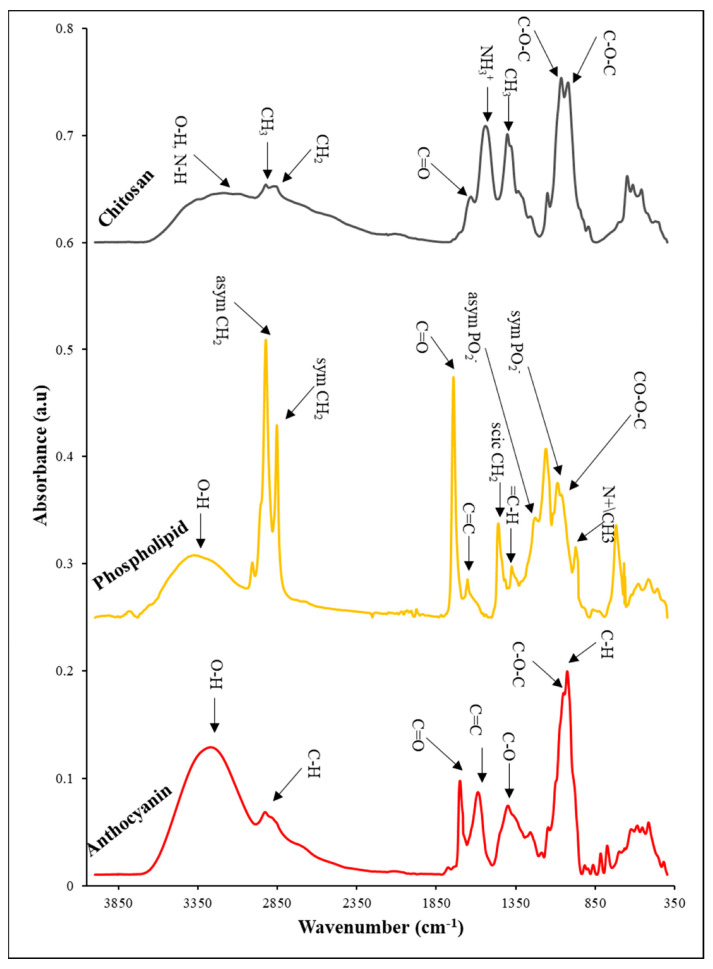
FTIR spectrum of anthocyanin, phospholipid, and chitosan.

**Figure 6 pharmaceutics-14-01622-f006:**
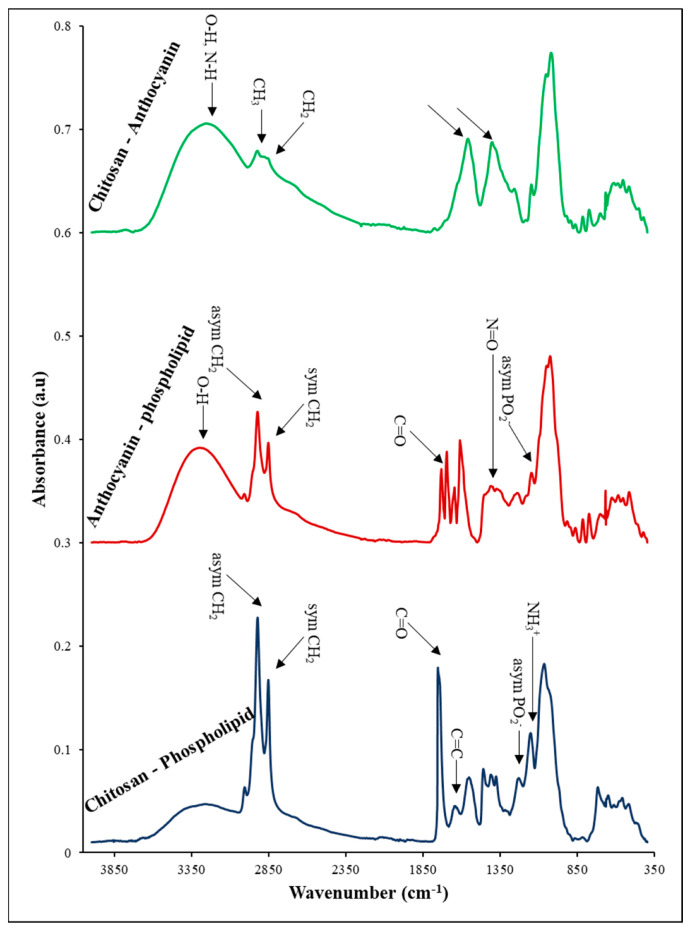
FTIR spectrum of chitosan-phospholipid, anthocyanin-phospholipid, anthocyanin-chitosan.

**Table 1 pharmaceutics-14-01622-t001:** Mean Particle size, PDI, zeta potential, and membrane fluidity of fabricated nanoliposomes.

Sample	Particle Size (nm)	PDI	Electrophoretic Mobility (µm·cm/V·S)	EE (%)
LP-FR-UH	121.60 ± 3.72 ^d^	0.198 ± 0.022 ^b^	−3.49 ± 0.11 ^c^	-
LP-AC-UH	132.41 ± 3.04 ^c^	0.224 ± 0.020 ^ab^	−3.26 ± 0.05 ^d^	42.57 ± 2.08 ^b^
LP-FR-CH	261.50 ± 8.69 ^a^	0.258 ± 0.017 ^a^	+4.51 ± 0.09 ^b^	-
LP-AC-CH	188.94 ± 6.15 ^b^	0.197 ± 0.014 ^b^	+4.80 ± 0.10 ^a^	61.15 ± 2.32 ^a^

All data are represented as mean ± SD. PDI: polydispersity index, EE: encapsulation efficacy. Different letters (a, b, c, d) reveal the significant differences (*p* < 0.05) between response variables.

**Table 2 pharmaceutics-14-01622-t002:** Membrane fluidity of various formulations of nanoliposomes.

Sample	Membrane Fluidity
LP-FR-UH	4.15 ± 0.07 ^a^
LP-AC-UH	3.41 ± 0.04 ^b^
LP-FR-CH	3.64 ± 0.02 ^c^
LP-AC-CH	1.39 ± 0.06 ^d^

All data are represented as mean ± SD. Different letters (a, b, c, d) reveal the significant differences (*p* < 0.05) between response variables.

**Table 3 pharmaceutics-14-01622-t003:** Rheological characteristics of various nanoliposome systems.

Sample	Power Low Model			Casson Model		
	k (Pa·s^n^)	n	R^2^	k_0C_ (Pa·s)	k_c_ (MPa·S)	R^2^
LP-FR-UH	0.001 ± 0.0002	1.007 ± 0.0932	0.9997	0.676 ± 0.036	0.961 ± 0.004	0.9949
LP-AC-UH	0.001 ± 0.0004	1.009 ± 0.0724	0.9998	0.529 ± 0.049	1.089 ± 0.025	0.9956
LP-FR-CH	0.018 ± 0.0005	0.899 ± 0.0319	0.9895	1.936 ± 0.009	9.025 ± 0.169	0.9846
LP-AC-CH	0.027 ± 0.0002	0.860 ± 0.0293	0.9832	2.304 ± 0.081	10.404 ± 0.289	0.9964

## Data Availability

The data presented in the current study are available upon the reasonable request to the corresponding authors.

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
