# Peer review of "Modifying the Stability and Surface Characteristic of Anthocyanin Compounds Incorporated in the Nanoliposome by Chitosan Biopolymer"

_pharmaceutics, 2022, doi:10.3390/pharmaceutics14081622_

Round 1

Reviewer 1 Report

The article emphasis on modifying the stability and surface characteristics of anthocyanin compounds incorporated in the nanoliposome by chitosan biolpolymer.

Abstract needs a conclusion statement in the end.

Introduction should be split in short paragraphs for better understanding of the readers

Heading 4 Discussion (Line 630) seems actually Conclusion rather than discussion Most of the discussion lacks proper reference. Proper citations are needed for the discussion section which is combined with the results section.   Equation numbers are mentioned in the text but not with the equation.

Author Response

The article emphasis on modifying the stability and surface characteristics of anthocyanin compounds incorporated in the nanoliposome by chitosan biopolymer.

  • Abstract needs a conclusion statement in the end.

Thank you so much for this precise point. We added the conclusion statement at the end of the abstract. please check it as highlighted on page 2; line no. 39-43

  • Introduction should be split in short paragraphs for better understanding of the readers

Based on the respective reviewer’s comment, the introduction was split into several paragraphs.

  • Heading 4 Discussion (Line 630) seems actually Conclusion rather than discussion.

Thank you for your proper comment. Section 4 was corrected as the “Conclusion” part and the overall conclusion of this study has been presented in this part. Please consider it as highlighted on pages 25-26, line nos. 664-690.

  • Most of the discussion lacks proper reference. Proper citations are needed for the discussion section which is combined with the results section.

 Thank you so much for this precise point. It was forgotten to cite the related reference to this section. We added the appropriate references to the mentioned section. Please check it as highlighted in the entire manuscript.

  • Equation numbers are mentioned in the text but not with the equation.

We thank the respective reviewer in advance for his/her proper comment. We added the equation number close to each equation. Please consider them as highlighted in line no. 221, 230, 246, 286, and 292

Reviewer 2 Report

The paper present the obtaining and the characterization of anthocyanin-loaded liposomes covered with chitosan. The presence of a chitosan layer was demonstrated, together with the improved mechanical stability. The interaction between the components are studied by FTIR spectroscopy. Some aspects should be improved.

The aim of the present work has to be better evidenced.

What is the improvement/novelty of the present system compared with other anthocyanin-loaded liposomes from the literature? (https://link.springer.com/article/10.1007/s13233-020-8039-7 , https://onlinelibrary.wiley.com/doi/full/10.1002/fsn3.2649 , etc.)

In order to estimate the utility of the LP-AC-CH, the release of anthocyanin or the antioxidant activity can be studied.

In the Introduction, some of the statements must be accompanied with references (Among various methods, the potential of liposomes as an encapsulation medium has 61 been proven in preserving, stabilizing, and delivering nutraceutical.  Nev-73 ertheless, despite all of the mentioned advantages, nanoliposomes have the problem of a 74 short half-life in the gastrointestinal tract.  This hydrophilic compound is biodegradable and bio-83 compatible, which has been of interest as mucoadhesive compound.)

Line 90: “as well as ” must be replaced with other word to understand the meaning of the sentence.

The number of the equations must be introduced.

Chapter 2.2.4: what is the ratio anthocyanin : lecithin and the ratio lecithin : chitosan?

Chapter 2.2.7: a reference must be introduced regarding the determination of AC content.

Chapter 2.2.10: The liposomes were dried for the FTIR spectroscopy?

Line 308: “attraction reactions must be replaced  with attraction forces” or something like that.

For the Figures 2 a and b, a better resolution must be provided. For a better understanding, a different color can be used for each sample.

Table 3: “K” can be replaced with “k” from eq. 4. The measurement unit of this parameter can be also introduced in the table.

Line 474: “Figure 3-d” must be replaced with “Figure 2-d”.

Line 517 : “saccharine” must be replace with “saccharide”.

Author Response

Reviewer #2

The paper present the obtaining and the characterization of anthocyanin-loaded liposomes covered with chitosan. The presence of a chitosan layer was demonstrated, together with the improved mechanical stability. The interaction between the components are studied by FTIR spectroscopy. Some aspects should be improved.

  • The aim of the present work has to be better evidenced.

Thank you for your precise comment. We explained the aim of this study more precisely in the introduction section. Please consider it as highlighted on pages 4 and 5, line nos. 122-135.

  • What is the improvement/novelty of the present system compared with other anthocyanin-loaded liposomes from the literature?

We are thanking the respective reviewer for this proper point. We briefly added some related studies and the novelty of the present study including employing a natural lecithin compound, omitting cholesterol in nanoliposome preparation, and applying a hydration method with the combination of ultrasonic and high-pressure homogenization techniques to establish stable a nanoliposome system coated with chitosan. Please check these corrections as highlighted in the manuscript (pages 4 and 5, line nos. 122-135)

  • In order to estimate the utility of the LP-AC-CH, the release of anthocyanin or the antioxidant activity can be studied.

Thank you for your precise comment. We also studied the antioxidant activity and release behavior of anthocyanin compounds from different nanoliposome systems in the various simulated gastrointestinal conditions. Such data is provided for another publication.

  • In the Introduction, some of the statements must be accompanied with references.

According to the respective reviewer comment, the related reference(s) was added to the mentioned section as below:

  • “Among various methods, the potential of liposomes as an encapsulation medium has been proven in preserving, stabilizing, and delivering nutraceuticals.”

Please check related reference(s) as highlighted on page 3; line nos. 89-90.

  • Nevertheless, despite all of the mentioned advantages, nanoliposomes have the problem of a 74 short half-life in the gastrointestinal tract.

Please check related reference(s) as highlighted on page 4; line nos. 102.

  • This hydrophilic compound is biodegradable and bio-83 compatible, which has been of interest as a mucoadhesive compound.

Please consider the added reference(s) as highlighted on page 4; line nos111.

  • Line 90: “as well as ” must be replaced with other word to understand the meaning of the sentence.

Considering the proper comment of the respective reviewer, the mentioned word was replaced with “and” and the statement was corrected as “Note that chitosan coating on nanoliposomes occurs through ionic interaction between amine groups with the positive charge in the chitosan and the diacetyl phosphate groups with the negative charge on the surface of liposomes”. Please check it as highlighted on page 4, line nos. 115-118.

  • The number of the equations must be introduced.

We thank the respective reviewer in advance for his/her proper comment. We added the equation number close to each equation. Please consider them as highlighted in line no. 221, 230, 246, 286, and 292

  • Chapter 2.2.4: what is the ratio anthocyanin : lecithin and the ratio lecithin : chitosan?

Thank you so much for this precise point. the ratio anthocyanin: lecithin and the ratio lecithin: chitosan was added to the 2.2.4 section as “It should be stated the ratio of anthocyanin extract to rapeseed lecithin as well as rapeseed lecithin to chitosan were 1.5:1 and 3:1, respectively”. Please consider it as highlighted on page 6, line nos. 184-186.

  • Chapter 2.2.7: a reference must be introduced regarding the determination of AC content.

According to the respective reviewer comment, the related reference was added to this section. Please consider it as highlighted on page 7, line nos. 221-222 and 231.

  • Chapter 2.2.10: The liposomes dried for the FTIR spectroscopy?

Liposome samples were lyophilized for the FTIR spectroscopy, we added this correction to section 2.2.10 as highlighted on page 9 line no. 272.

  • Line 308: “attraction reactions” must be replaced with “attraction forces” or something like that.

We are thankful for your valuable comments. The “attraction reactions” was replaced with “attraction forces”. Please consider it as highlighted on page 11 line no. 354.

  • For the Figures 2 a and b, a better resolution must be provided. For a better understanding, a different color can be used for each sample.

Thank you so much for this valuable point. For providing better resolution, figure 2 was split into 2 separated figures. Figures 2a and 2b were regarded as Figure 2 and figures 2c and 2d were considered as Figure 3a and 3b and all following figures were corrected as well.  

  • Table 3: “K” can be replaced with “k” from eq. 4. The measurement unit of this parameter can be also introduced in the table.

Thank you so much for this precise point. We corrected “K” as “k” and its unit was introduced in the table. Please consider them as highlighted in table 3, page 16.

  • Line 474: “Figure 3-d” must be replaced with “Figure 2-d”.

Thank you so much for this precise point.  The number of this figure was changed as mentioned in the two previous comments.

  • Line 517 : “saccharine” must be replace with “saccharide”.

According to the respective reviewer's comment, the mentioned phrase was replaced with the proposed one by the respective reviewer. Thank you for your appropriate comment. Please check it as highlighted on page 19; line nos. 569.

Round 2

Reviewer 2 Report

The paper was improved. The autors have considered the suggestions.